

# A Promising Downscaling Strategy for Topographic Heavy Rainfalls over the Asian-Australian Monsoon Region by Leveraging Multi-Scale Moisture Dynamical Control

Jia-Xin Guo[1], Wei-Ting Chen[1], Chien-Ming Wu[1]

[1]Department of Atmospheric Sciences, National Taiwan University, Taipei, 106319, Taiwan

*Correspondence to*: Wei-Ting Chen (weitingc@ntu.edu.tw)

**Abstract.** Conventional downscaling methods are severely challenged in providing future projections for local precipitation extremes over tropical areas with complex terrain. Here, we investigate the multi-scale moisture dynamical control on the topographic heavy rainfalls over Taiwan to construct an alternative downscaling pathway based on crucial physical

processes. We identify large-scale regimes of vertically-integrated vapor transport (IVT) with objective machine learning classification and then quantify embedded upstream IVT conditions that drive distinct local rainfall responses over Taiwan, which are verified by semi-realistic large-eddy simulations with realistic topography. The identified environmental control and the resolved convective processes form the physical basis for providing interpretable and high-resolution rainfall projections in a changing climate. In addition to Taiwan, the applicability of this downscaling strategy to four areas in the

Asian-Australian monsoon region exhibiting high sensitivity of heavy topographic rainfall to the upstream moisture dynamics is also discussed.

## 1 Introduction

Most of the rainiest places in the summer Asian-Australian monsoon (AAM) region lie around complex topography ranging from coast to mountains (Chang et al., 2005; Xie et al., 2006; Biasutti et al., 2012; Shige et al., 2017). How these

topographic heavy rainfalls might manifest in a warming climate is a central problem in both social and scientific sectors. Answering this critical question requires downscaling future projections from global climate models (GCMs) to refined spatial scales. Yet conventional downscaling approach is prone to producing ill-informed results due to the uncertain response of regional circulation in GCM ensembles and the limitations of downscaling models. A process-based alternative pathway is therefore in urgent need to provide decision-relevant information and interpretable future projections (Hazeleger

et al., 2015; Maraun et al., 2017; Shepherd et al., 2018).

The first layer of interpretability stems from addressing the environmental control on topographic heavy rainfalls, which includes large-scale influences from regional circulation states and the dominant forcing types near the targeted area. Regional circulation responses are generally uncertain due to internal variabilities and imperfect model representations (Shepherd, 2014; 2019), a predicament further complicated by the prominent multi-scale interaction within the AAM



regional circulation. Hence, instead of downscaling unconditionally from GCM-projected large-scale states, focusing on the better-known thermodynamic impacts of climate warming on local weather—with the relevant large-scale states conditioned—is argued to provide more informative assessments (Trenberth et al., 2015; Shepherd, 2016; Maraun et al., 2017). This in turn requires an objective classification of large-scale regimes and a subsequent identification of the ones that promote heavy rainfalls. Additionally, large-scale regimes often do not hold enough discriminative power to detailed local

heavy rainfall outcomes (Plaut et al., 2001; Boé et al., 2006), where embedded flow variations also exert substantial control (Pasquier et al., 2019). This is especially true in the AAM region, where the near-coast topographies can serve as both dynamical barriers and diurnal heating sources during summer (Xie et al., 2006; Biasutti et al., 2012; Shige et al., 2017). The upstream flow condition is thus a crucial environmental component indicating the dominant forcing for a given rainfall event. The next layer of interpretability lies in how the fine-scaled rainfall outcomes are rendered through modelling tools. The

convection developed over complex topography involves numerous critical processes such as buoyancy differences, land-sea breezes, and orographic gravity waves (Kirchbaum et al., 2018). A model capable of resolving these crucial processes can support credible results and mechanism investigation for physical understanding. Regional climate models (RCMs) conventionally used in dynamical downscaling remain challenged in this respect with concerns about resolution, parametrized physics, and the representation of topography (Di Luca et al., 2015; Giorgi, 2019). Moreover, given the pivotal

role of environment–topography interaction in shaping the heavy rainfall variabilities, we argue that a simplified representation of the forcing environment can highlight the interactive process with topography and its resultant influences on rainfall outcomes (Kirshbaum and Smith, 2009; Kirchbaum, 2011; Metzger et al., 2014; Panosetti et al., 2016), and, hence, the interpretability of the projected results.

To this end, we use topographic heavy rainfalls over Taiwan (Fig. 1) to investigate such a process-based downscaling

framework. The complex topography of Taiwan features abrupt gradients over short horizontal distances, resulting in complicated heavy rainfall behaviors during summer, closely tied to both large-scale circulation patterns and upstream flow variations. To focus on the moisture dynamical influences from the environment, we examine the large-scale patterns and the upstream flow using the low-level vertically-integrated vapor transport (IVT) (Fig. 1a–b). In particular, a machine-learning-based classification is adopted to objectively establish daily large-scale IVT regimes (Fig. 1a), considering that multi-scale

spatial features such as the monsoon circulation and various moist disturbances are difficult to capture faithfully with linear-based methods. Furthermore, in order to highlight the environment–topography interaction as previously mentioned, we use a large-eddy simulation model equipped with realistic Taiwan topography (TaiwanVVM, Wu et al., 2019) and initiate simulations with a profile representative of the dominant environmental forcing, instead of using weather forecast models with nested domains. This is referred to as the "semi-realistic" simulation approach (more in Methods). Successful

simulation of past events will verify the effectiveness of this framework and provide foundation for projecting the future. Although Taiwan is often missed in satellite-based heavy rainfall investigations (e.g., Xie et al., 2006; Nicholas and Boos, 2024) due to limitations of detecting the precipitation features in satellite products, posed by the complex terrain (Huang et al., 2018; Pradhan et al., 2022, and Fig. S1), the urgency of a process-based downscaling strategy for topographic heavy



rainfalls is shared with many areas in the AAM region. Therefore, the objectives of this study are (1) to objectively establish
large-scale circulation states of the AAM region via IVT, (2) to identify the large-scale and upstream moisture dynamic
environments critical to the complicated heavy rainfall behaviours over Taiwan, and (3) to adopt large-eddy simulations with
realistic topography to verify the physical pathway. The data and modelling tools for fulfilling these objectives are described
in Section 2 with results presented in Section 3. In section 4, we present mechanism discussion with other studies and also
preliminary applications to the future climate and other topographically-complex areas in the AAM region.

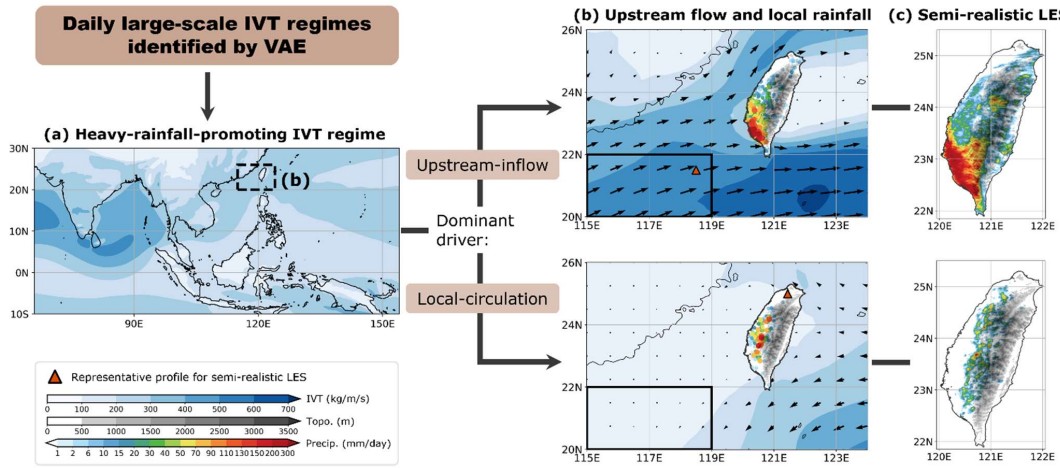

**Figure 1: Schematic diagram illustrating the multi-scale moisture dynamical control on Taiwan topographic heavy rainfalls. (a)**
**Example of the large-scale IVT intensity pattern over the AAM region (15° S – 30° N and 66 – 154° E) identified to bear higher**
**potential for heavy rainfalls over Taiwan (see Section 3.1). Dashed-lined box indicates the region of (b). (b) The IVT environment**
**surrounding and the rain gauge daily rainfall over Taiwan. Arrows represent the IVT vectors. Solid-lined box (20 – 22° N, 115 –**
**119° E) is the upstream region defined for Taiwan in this study. (c) Semi-realistic large-eddy simulations from TaiwanVVM.**
**Upper (lower) panels in (b)(c) are cases from August 10, 2014 (August 19, 2012) exemplifying different upstream conditions (see**
**Sections 3.2 and 3.3).**

## 2 Methods

### 2.1 Data

We focus on boreal summer (April to September) to study the AAM domain (15° S – 30° N and 66 – 154° E, Fig. 1a), and
the region over northern South China Sea (20 – 22° N, 115 – 119° E, solid-line box in Fig. 1b) is additionally targeted as the
upstream region of Taiwan. Several datasets are used to investigate the moisture environmental control on heavy rainfalls
with focus on certain areas and periods.

We calculate the domain-wide low-level vertically-integrated vapor transport (IVT) field to examine the moisture
environment. IVT is defined following Eq. (1):



$$IVT = -\frac{1}{g}\int_{1000}^{700} q \cdot \vec{v}\, dP, \tag{1}$$

where $g$ is the gravitational constant, $q$ is specific humidity, and $v$ is the wind vector. IVT calculated from ECMWF Reanalysis v5 (ERA5) (Hersbach et al., 2020; 2023) daily data in 2001–2019 with 0.25° spatial resolution are used to identify large-scale regimes and conduct observational investigation.

To demonstrate the applications of our proposed strategy with GCMs, we additionally use IVT calculated from Taiwan Earth System Model Version 1 (TaiESM1) (Lee et al., 2020)—a CMIP6 GCM participant with leading ability to simulate Asian and Western North Pacific monsoon (Wang et al., 2021)—in 1.25° (0.94°) spatial resolution in the zonal (meridional) direction and daily time scales. TaiESM1 historical data in 2000–2014 and future projection under SSP5-8.5 scenario in 2080–2100 are used in this study.

Heavy rainfall is identified with daily accumulated precipitation. For Taiwan, we use the hourly rain gauge data from the Central Weather Administration (CWA) to calculate the daily accumulation. The results are presented with focus on both the southwestern area (21.9 – 23.5° N, 120 – 121° E) and the entire island, which are covered by 233 and 740 rain gauges, respectively, with slight variations in the number of stations throughout the study period that do not affect the presented results. For other parts of the AAM region, we use the IMERG (Integrated Multi-satellitE Retrievals for Global Precipitation

Measurement) V07B 0.5-hourly precipitation data with 0.1° spatial resolution (Huffman et al., 2023). We note that the daily rainfall over Taiwan is obtained based on the local time frame (UTC+8) to compare with the full-day TaiwanVVM simulations initiated at local midnight, while the ERA5 and IMERG data are based on UTC. Rainfall datasets are also used for 2001–2019.

Additional datasets used in this study includes the ERA5-Land (Muñoz Sabater et al., 2019) time-invariant surface

geopotential of 0.1° spatial resolution for the topographic elevation plotted in Fig. 8a and the International Best Track Archive for Climate Stewardship (IBTrACS) data (Knapp et al., 2010; Gahtan et al., 2024) for presenting the tropical cyclone positions (when documented as "tropical storms" in the dataset) in the Supplement Fig. S2b.

## 2.2 Interpretable machine learning with Variational Autoencoder

To effectively characterize large-scale IVT regimes in the AAM region, we adopt a data-driven classification based on

Variational Autoencoder (VAE) (Kingma and Welling, 2013; Rezende et al., 2014). VAE is a machine learning network built on the concept of dimensionality reduction (or manifold hypotheses), which argues a high-dimensional data space can be represented by a much smaller number of features. Advantages of VAE in extracting regimes from daily patterns include the "variational inference" via a given prior distribution during backward propagation, which regularizes the low-dimensional data space (latent space), and the information bottleneck that forces VAE to reserve key information in the input

data (Alemi et al., 2016). Consequently, VAE can faithfully represent original inputs using a limited number of features and arrange them reasonably in the latent space.





In practice, VAE maps input samples to dimension-reduced forms with an encoder and strives to reconstruct them back into original forms with a decoder. Figure 2 shows the structural details of the VAE deployed in this study, consisting of a nearly symmetrical pair of an encoder and a decoder, connected by a two-dimensional latent space. The convolutional layers with
nonlinear activation function of the encoder are advantageous in extracting generalizable structures and regional differences, which makes VAE more suitable for representing the multi-scale features of AAM circulation and generating regimes than linear-based method such as the empirical orthogonal function (EOF) (see Fig. S4 for examples). Moreover, using a multivariate Gaussian distribution as the prior distribution for dimensionality reduction, our VAE possesses a continuous latent space, on which the IVT pattern features transition smoothly (Fig. 3). In addition to the encoded vectors from the input
samples, the continuous latent space allows the decoder to generate reasonable reconstructions from arbitrary latent vectors after successful learning. This provides extensibility to unseen datasets such as from GCMs and future periods, which is a huge advantage over other clustering methods.

We train our VAE with summer IVT intensity images from ERA5 over the AAM region. The 3477 daily samples from 2001–2019 are further split into three subsets for proper model training: First 2000 days as the training set, the subsequent
500 days as the validation set, and the last 977 days as the testing set. Scores of pattern correlation and root mean square error (Fig. S5) and the composite of each large-scale IVT regime from the testing set samples (Fig. S3) suggest our VAE can encode unseen IVT samples and reconstruct them with reasonable outcomes. As one of our focuses is to find the relationships that these large-scale IVT regimes hold with local precipitation, and that the precipitation data is not used during the VAE training process, results shown in this study are based on the entire data period of 2001–2019.

The input IVT fields have gone through two steps of pre-processing: individually normalized by the daily maximum and minimum over the field following Eq. (2) and coarse-grained from the original 0.25° spatial resolution to 1.25° by skip-sampling. The individual normalization allows the spatial structure of each day to be independently emphasized, and the resolution adjustment can effectively trim down the number of VAE parameters, which improves the speed of convergence and reduces computational load. An additional advantage of being successfully trained on coarser resolution IVT patterns is
that the encoder is readily applicable to recognizing IVT fields from GCMs, which benefits GCM model evaluation and future projection (see Fig. 7 and Discussion).

$$I_i' = \frac{I_i - I_{min}}{I_{max} - I_{min}}, \tag{2}$$

$I$ represents the original IVT intensity, primed after scaling. The subscript $i$ stands for the $i^{th}$ grid, whereas the subscript $min$ and $max$ stand for the minimum and maximum taken over that daily field.





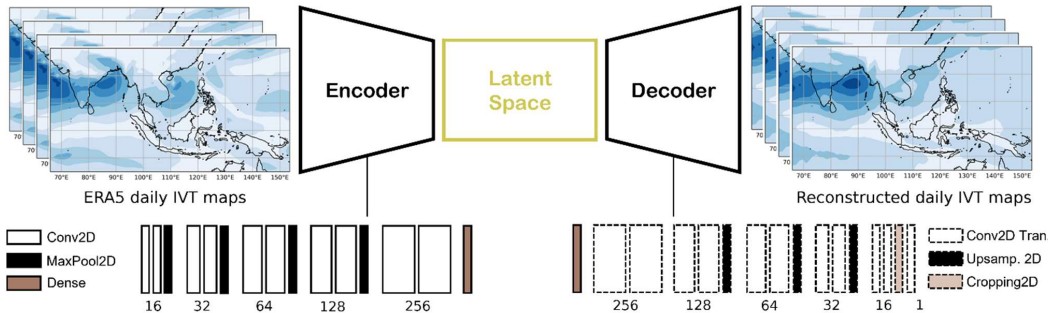

**Figure 2: Structure of the Variational Autoencoder used in this study. The boxes in the bottom row denote the neural network layers, with the colours representing the layer types (denoted on both ends) and numbers representing the channels.**

### 2.3 Semi-realistic large-eddy simulation with TaiwanVVM

TaiwanVVM is a large-eddy simulation (LES) model based on the Vector Vorticity Equation Cloud-Resolving Model (VVM) (Jung and Arakawa, 2008). VVM utilizes a distinctive dynamical core based on the anelastic vector vorticity equation, which excels at resolving vorticity features and by extension the buoyancy and convective processes. Over the years, VVM has evolved to encompass idealized block topography (Wu and Arakawa, 2011; Chien and Wu, 2016) and the realistic complex terrain over Taiwan (TaiwanVVM) (Wu et al., 2019) and couple with the Noah Land Surface Model (Chen and Dudhia, 2001), which allow idealized and realistic experiments to investigate detailed physical processes as well as their real-world impacts. Additionally, VVM adopts the Predicted Particle Properties (P3) microphysics scheme (Morrison and Milbrandt, 2015; Huang and Wu, 2020) and is able to produce convection and precipitation intensities similar to observed events. The high spatial resolution of 500 m is the most widely-used in the past TaiwanVVM studies (e.g., Chang et al., 2021; Hsieh et al., 2022; Chang et al., 2023; Hsu et al., 2023; Chen et al., 2024a) and is adopted in this work, when even finer resolution is available for idealized simulations in VVM (such as 100 m in Wang et al., 2024). Equipped with adequate cloud-resolving abilities, land surface processes with realistic topography, and microphysics, TaiwanVVM is the ideal tool for investigating convection–land interactions over one of the most topographic-challenging places like Taiwan.

As a LES model, TaiwanVVM simulations are initiated by applying an environmental profile uniformly to all grid columns across the simulation domain and let the physical processes evolve freely over the topography. This approach is well-suited for places where topography is complex and that the environment–topography interaction largely shapes the local weather. We refer to this as the "semi-realistic" approach, since the profiles of temperature, specific humidity, and horizontal winds are taken from observational sounding or reanalysis data that best represents the dominant environmental forcing. The TaiwanVVM simulations in this work are performed on a domain spanning 512 km in zonal and meridional directions with 500 m horizontal resolution, where Taiwan is placed at the center, and 100 m vertical resolution with stretching grids in the



upper portion. All cases are simulated from 00 LST for 24 hours with 10-second integration time step and 10-minute output
frequency. The initial settings are documented in Section 3.3 within relevant contexts.

## 3 Results

### 3.1 Large-scale moisture dynamical regimes identified by the Variational Autoencoder

We first establish large-scale moisture dynamical regimes using domain-wide IVT intensity patterns to characterize distinct
regional circulation states. This is achieved by interpreting the clustered samples in the latent space of VAE after successful
training. The use of normalized IVT intensity fields helps constrain the number of identified regimes, and adopting an
objective data-driven approach provides a more physically-consistent foundation for further analyses. The direction and
original intensity of IVT are later examined in the local upstream (Section 3.2).

The latent space of our VAE is two-dimensional, which best supports the visualization and interpretation of encoded
information than higher-dimension latent spaces (Chen et al., 2024b; Hsieh et al., 2024). Figure 3a shows how IVT intensity
fields of the entire data period are encoded into the two-dimensional latent space. We target the square of [-6,6] along each
latent dimension, which contains 99.5% of all samples, and delineate 36 unit-square subspaces (Fig. 3b). Each subspace
constitutes a large-scale IVT regime, and their distinct flow features are unveiled through the decoder-reconstructed IVT
patterns from the midpoints (Fig. 3c). We regard Fig. 3c as representatives of each regime, whose credibility is ensured by
comparing the sample composites retrieved from the corresponding regimes (Fig. S2). Please be noted that, as we emphasize
the daily spatial structures by individually normalizing the training IVT fields, the representative IVT patterns illustrate the
relative strength of weather systems. In other words, it is possible for IVT patterns in the same regime to exhibit different
intensities (see Fig. S6 for examples).

Based on visual inspection of the daily samples and the representative IVT patterns in Fig. 3c, the regimes are grouped into
three categories (A, B, and C) (Fig. 3b–c), with each pattern therein bearing some minutiae and can be regarded as a subtype.
We show in Fig. S7 that the AAM IVT patterns transition through the three categories throughout the boreal summer
generally month-to-month, whereas day-to-day variations mostly occur within categories. Primary characteristics of each
category are summarized as follows: Category A (mainly emerge in Apr, May, and Sep) features stronger IVT over the
Western North Pacific and the Northern Indian Ocean. The former mainly results from tropical easterlies, sometimes
extending westward to the Southern China Sea, while the latter is most often associated with the developing cross-equatorial
flow over the Indian Ocean, sometimes with tropical disturbances embedded. Category B (Jun–Sep) features prevailing
South Asian Monsoon flow across the boreal subtropics, resulting in pronouncedly strong IVT on both sides of the Indian
Peninsula. From left to right in Category B, the extended strong IVT—mostly from strong monsoon flow or tropical
disturbances—goes further eastward. Finally, Category C (Jul–Sep) features an exclusively high percentage of days with
tropical cyclone incidences, especially distributed across the northern South China Sea and east off Taiwan (see also Fig.
S2b). In comparison to the Category B patterns also with tropical cyclone occurrences (Fig. S2b), Category C patterns





usually feature retreating South Asian Monsoon circulations and are thus more frequently found in late summer. Still, the enhanced flow southward to the tropical cyclone systems can be greatly hazardous to its passing regions (Bagtasa, 2017; 2019). Note that the relatively weak colour of Category C patterns in Fig. 3c results from the concentrated strong IVT of tropical cyclone cores after normalization (see Fig. S8 for examples) and the positional variability of these IVT cores from

case to case. Despite the differences in reconstructed intensity across categories, the multi-scale features across the domain are sufficiently preserved for regime classification.

This flow-pattern-based, daily classification adds more details upon the seasonal- or sub-seasonal averaging approach commonly adopted in the AAM monsoon rainfall research, facilitating future climate applications where seasonal cycles may shift. It has roots in the weather-typing method from statistical downscaling and shares conceptual similarities with the

circulation-analogue methods to find analogous large-scale dynamical states, more widely-used in mid-latitude studies (e.g., Yiou et al., 2007; Cattiaux et al., 2010). Our method further improves upon them as the continuous latent space better handles the "intra-type" problem (Wilby et al., 2004), and the encoder classifies samples based on multi-scale structural similarities instead of linear pattern correlation. These large-scale IVT regimes serve as the first conditioning basis to explore downstream local rainfall responses.



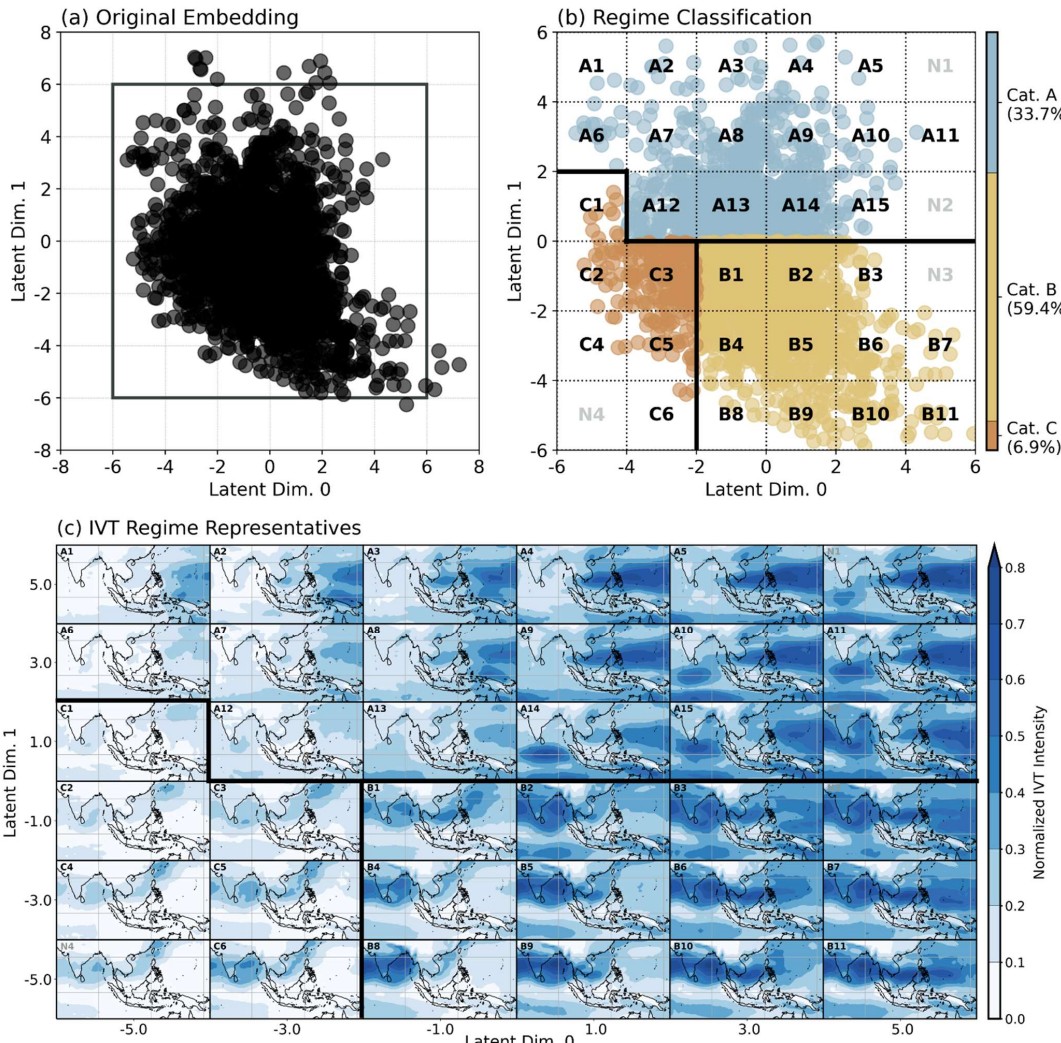

**Figure 3: Two-dimensional latent space visualization. (a) The original latent codes of all samples based on large-scale IVT intensity patterns. (b) The zoom-in of the square region in (a), further divided (dashed lines) into 36 subspaces (regimes). (c) IVT patterns representing the 36 large-scale IVT regimes, decoded from the midpoints of subspaces in (b). Three categories are delineated by thick black lines in (b)(c) and marked by colours in (b). Text in the middle of each subspace of (b) and the upper left of each IVT pattern of (c) denote the regime names, where grey ones indicate empty regimes.**



**3.2 Multi-scale moisture dynamical control on Taiwan topographic heavy rainfalls from observation**

After identifying representative large-scale IVT regimes over the AAM region (Fig. 3), we now move on to the heavy rainfall outcomes over Taiwan with respect to both large-scale states and upstream flow variations. Common practice of investigating regime-dependent rainfall behaviours is to derive rainfall composites conditioned on the identified regimes.

Here, we emphasize each past event as an opportunity for investigating the environmental control (Shepherd et al., 2018) and thus carry out the following rainfall analyses on a daily event basis.

The rainiest part of Taiwan during summer is the southwestern area (21.9 – 23.5°N, 120 – 121°E), resulting from both highly-localized mountainous convection and widespread heavy rainfall events (Johnson and Bresch, 1991; Chen et al., 1999; Wang et al., 2008), owing to the active afternoon thunderstorm driven by surface heating and the strenuous mountain slopes

windward of the prevailing low-level southwesterly. Remarkable rainfall intensities can be achieved by both rainfall patterns, which bear high impacts to local areas (Chen et al., 1999; Tu et al., 2014; Chang et al., 2021; Chang et al., 2023; Chen et al., 2024a). We define two rainfall metrics to characterize these varied heavy rainfall behaviours over southwestern Taiwan: (a) the relative daily rainfall ratio ($R$) following Eq. (3):

$$R = \frac{S\ (southwestern\ Taiwan)}{S\ (Taiwan)}, \tag{3}$$

where $S$ represents the *area-summed* daily rainfall, and (b) the *area-average* daily rainfall ($A$) following Eq. (4):

$$A = \frac{S\ (southwestern\ Taiwan)}{\#\ of\ rain\ gauges\ (southwester\ \ Taiwan)}. \tag{4}$$

Southwestern Taiwan is likely to experience localized rainfall patterns on days with high relative ratio $R$ but low area-average $A$.

Figure 4a–b illustrates the relationship between large-scale IVT regimes and southwestern Taiwan rainfall by colouring the

encoded daily samples with these two rainfall metrics. Both metrics frequently peak under six of the large-scale IVT regimes (C3, C5, B1, B2, B4, B5) that feature relatively strong South Asian Monsoon circulation or nearby tropical cyclones in the Western North Pacific (Fig. 3c). The distribution of higher values in Fig. 4a–b suggests that, across all identified large-scale states in summer, intense precipitation events over southwestern Taiwan are most promoted under the six marked large-scale IVT configurations.

Discriminative rainfall patterns seem to further depend on the upstream flow variations embedded in the six large-scale IVT regimes associated with heavy rainfall over southwestern Taiwan. Conditioned on the six large-scale IVT regimes, we distribute the daily samples according to the upstream IVT intensity and direction and colour them with the rainfall metrics (Fig. 4c–d). The "upstream region" for southwestern Taiwan is defined as a northeastern area of the South China Sea (20 – 22° N, 115 – 119° E, Fig. 1b) to examine the incoming environmental flow while minimizing impacts from coastal

mesoscale convective systems (MCSs). We use the average over this upstream region to represent upstream IVT. Figure 4c shows that high relative ratio ($R$) of southwestern Taiwan rainfall distributes over weak upstream IVT and strong southwesterly IVT, while Fig. 4d shows that high area-average ($A$) of southwestern Taiwan rainfall occurs mainly under





strong (≥250 kg/m/s) southwesterly to westerly IVT (200°–280°). Consequently, Fig. 4c–d reveals differed heavy rainfall patterns as a response to the upstream flow variation: an "upstream-inflow-dominant" rainfall type driven by moist and
strong upstream inflow and resulting in higher area-average rainfall amount $A$ (the samples in the marked sector of Fig. 4c–d), and a "local-circulation-dominant" rainfall type either with weak moist inflow or without moist flow directing towards inland, which tends to have more localized rainfall patterns (lower area-average $A$, Fig. 4d).

Figures 5 and 6 illustrate individual events and all-event composites, respectively, of the two rainfall types defined in Fig. 4d. Please be noted that the precipitation intensity scales in Fig. 5 are identical for all cases. When the upstream moist inflow is
weak or the upstream flow does not direct towards Taiwan, the local circulation by driven surface heating is the dominant driver of intense rainfalls over land. The intense rainfalls emerge along foothills (~500 m elevation) as clustered hotspots (Fig. 5a–b and 6b) and some randomly over the island. The rainfall features are consistent with past studies documenting Taiwan rainfall behaviours under weak southwesterly flow and the subsidence of the subtropical high (Johnson and Bresch, 1991; Chang et al., 2021; Chang et al., 2023; Chen et al., 2024a). In contrast, when the upstream moist inflow is strong, it
becomes the dominant driver triggering widespread heavy rainfalls from coastlines to mountain slopes over the southwestern area (Fig. 5c–e), and the precipitation pattern generally forms a northward gradient (Fig. 6e). With the prominent moisture supply, rainfall accumulates more often to disastrous amounts (Fig. 5c–e). We highlight that the directional and intensity thresholds of incoming flow separating the two rainfall types (Fig. 4d) are greatly determined by the local topographic features. More about the transition of dominant drivers will be discussed in Section 4.3.

We show in this subsection that summer heavy rainfalls over Taiwan emerge more frequently under six large-scale IVT regimes (Fig. 4a–b), and upstream moisture transport conditions reveal the dominant drivers of distinct rainfall patterns over the island (Fig. 4c–d, 5, and 6b, e). The additional analyses of embedded upstream IVT conditions allow us to delve efficiently into the individuality of past events up to a very fine scale. It is more usual for downscaling studies to utilize either the large-scale circulation states or more adjacent regional environments, while here we showcase the promising
potential of adopting the multi-scale perspective of moisture dynamics. The pinpointed large-scale IVT regimes and upstream IVT conditions will serve as very useful bases for exploring plausible local-circulation-dominant and upstream-inflow-dominant events, in response to future moisture and wind conditions.



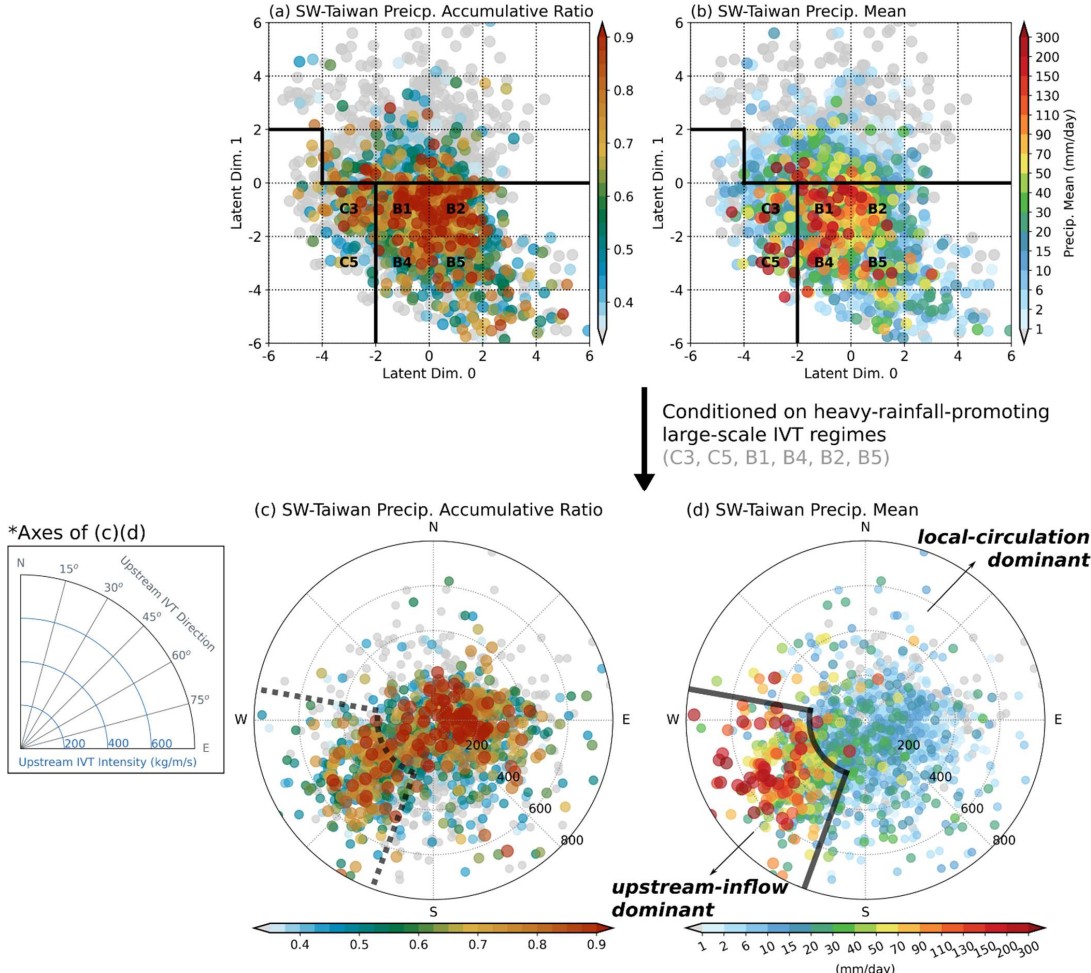

**Figure 4: Two metrics of southwestern Taiwan rainfall coloured against large-scale and upstream IVT conditions. Daily samples are arranged with respect to large-scale IVT patterns in (a)(b) and to upstream mean IVT conditions in (c)(d). A diagram explaining the axes in (c)(d) is provided with an inset plot. Colours in (a)(c) represent the relative daily rainfall ratio (R) over southwestern Taiwan relative to the island-wide total. Colours in (b)(d) represent the area-average daily rainfall (A) over southwestern Taiwan. The dashed-line in (c) and the solid line in (d) mark the upstream IVT condition where most of the severe rainfalls take place, visually defined from (d). Note that (c)(d) only contain samples within the marked large-scale IVT regimes (C3, C5, B1, B2, B4, B5).**



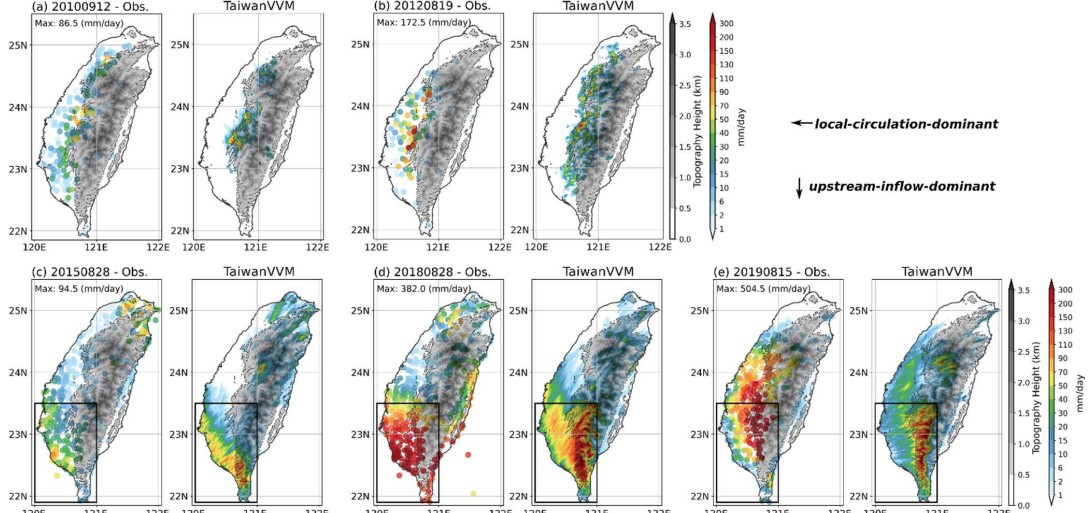

**Figure 5: Observed and TaiwanVVM-simulated event variability in heavy rainfall patterns over Taiwan. Local-circulation-dominant (upstream-inflow-dominant) cases are placed in the top (bottom) row. Left (right) panel of each subplot shows the observed (TaiwanVVM-simulated) pattern. Solid-line boxes in (c)–(e) mark the southwestern area. Maximum observed rainfall denoted in the left panels of all subplots are taken from the entire island for (a)(b) and from the southwestern area for (c)–(e).**

### 3.3 TaiwanVVM large-eddy simulations of local heavy rainfall events

To establish the physical pathways of the observational relationships, individual events in Fig. 5 and composite behaviors in Fig. 6 of the two rainfall types are simulated with the semi-realistic LES using TaiwanVVM (Wu et al., 2019). We take initial profiles at an upstream point (21.5° N, 118.5° E, triangle in the upper panel of Fig. 1b) from ERA5 to represent the upstream-inflow-dominant environment and those from the CWA operational sounding in Banqiao station at northern Taiwan (triangle in the lower panel of Fig. 1b) to represent the local-circulation-dominant environment. The wind fields in the upstream-inflow-dominant simulations are additionally nudged to initial conditions to maintain the impinging flow with a relaxation time scale of six hours. The sea surface temperature is prescribed at 29°C to highlight the sensitivity to initial atmospheric conditions.

Right panels of Fig. 5 show that TaiwanVVM well simulates the spatial and intensity features of past heavy rainfall events of both types, including the orographically-locked hotspots of the local-circulation-dominant events (Chang et al., 2021; Chang et al., 2023; Chen et al., 2024a) (Fig. 5a–b) and the spatial variability of the upstream-inflow-dominant events (Tu et al., 2014) (Fig. 5c–e and Fig. S9 for an additional demonstration). Moreover, choosing ~30 cases each from the two rainfall types to cover the upstream IVT conditions (Fig. 6a, d), the simulation composites (Fig. 6c, f) also closely match the observation (Fig. 6b, e) in spatial gradient, hotspot locations, and rainfall intensity—the key features of the heavy rainfall patterns. These successful numerical simulations confirm that the multi-scale moisture dynamical control is effective in



representing the key environments of the two distinct rainfall behaviours over Taiwan, especially when the model explicitly simulates the detailed physical processes within the environment¬–topography interaction for producing topographic heavy rainfall over complex terrain.

To this point, we exemplify a physically-grounded downscaling framework that provides the conditional bases of key moisture dynamical environments and the ability to simulate resultant topographic heavy rainfalls—including the spatial and intensity variabilities, in high resolution. This bears relevant implications for flood-risk measurements for residents and different industries in the plain and mountainous areas especially over southwestern Taiwan. Although TaiwanVVM is a model tailored for Taiwan's topography, high-resolution realistic modelling is essential as the final component of the

downscaling strategy to link environmental drivers to regional precipitation impacts over complex terrain (Schaller et al., 2020; Schemann et al., 2020). Moreover, the VAE-identified large-scale IVT regimes are readily applicable for areas within the AAM region, and the upstream IVT analyses can be easily done for other targeted local areas. Building on these results, we will next discuss ways of expanding this downscaling strategy into future topographic heavy rainfall projections and other AAM regions.





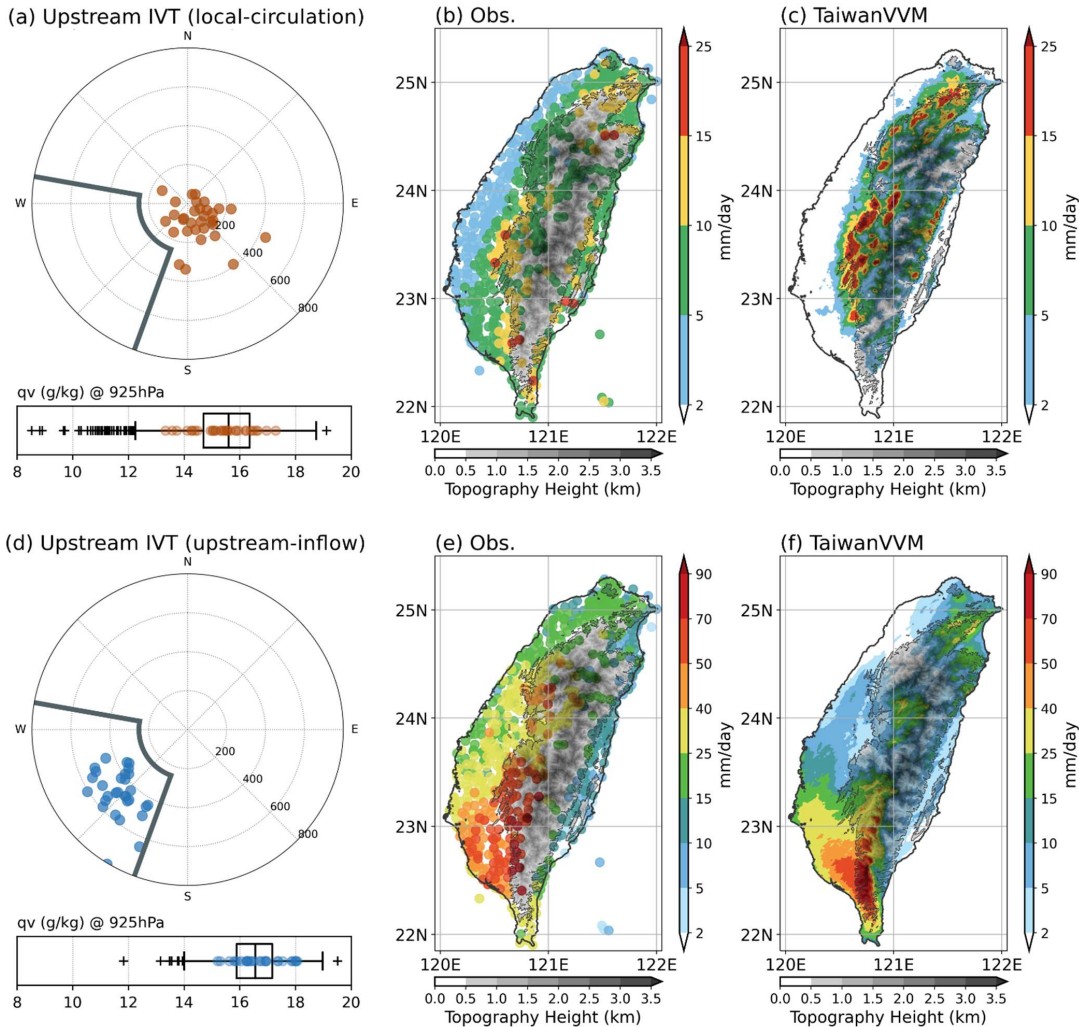

**Figure 6: Upstream conditions and rainfall patterns of the local-circulation and upstream-inflow dominant heavy rainfall types over Taiwan. Top (bottom) row shows the local-circulation (upstream-inflow) dominant rainfall types. (a)(d) The upper panels illustrate the upstream mean IVT conditions of the simulated cases. The lower panels show the distribution of specific humidity at 925 hPa at an upstream spot (21.5° N, 118.5° E, triangle in the upper panel of Fig. 1b) of all local-circulation/upstream-inflow dominant cases (box plot), and the simulated cases (coloured dots). (b)(e) Observational rainfall composites (all cases) from the CWA rain gauge data. (c)(f) Simulated rainfall composites (~30 cases) from TaiwanVVM.**



## 4 Discussion

### 4.1 Towards future projections of topographic heavy rainfalls

In view of the more uncertain responses of the Earth system's dynamical aspects than thermodynamical ones in a warming
climate (Shepherd, 2014), unconditional probabilities through GCM ensembles render less ideal in communicating risks and
impacts. In process-based pathways, conversely, future projections can now be described in terms of changes in the
thermodynamic and dynamical drivers. This allows various "conditioned" futures to be explored, which are termed
"storylines" of the future (Hazeleger et al., 2015; Shepherd et al., 2018).

Informative projections of local weather can be derived by assuming no apparent changes in the dynamical aspects and
focusing on the thermodynamic influences (Trenberth et al., 2015; Shepherd, 2016). Local precipitation over Taiwan can be
realized efficiently through adjusting the moisture profile with respect to a certain warming extent and running TaiwanVVM
simulations for individual events (such as Fig. 5) or rainfall types (such as Fig. 6), with the wind structure and regional
circulation states conditioned. Storylines of this sort aim for investigating plausible changes of weather events in magnitude,
hotspot locations, and event duration that facilitates inter-discipline communications in the decision space (Shepherd et al.,
2018; Sillmann et al., 2021). A +3K pseudo-global warming experiment conducted on local-circulation-dominant extreme
convection over Taiwan is an illustrative example (Chen et al., 2024a), where thermodynamic influences are found to
translate into increased convective intensity, enhanced duration variabilities, and expanded emergences towards the plain
area. As these hotspots are around several major reservoirs, this storyline bears implications to water resource management
in warmer summers.

Although exploring the dynamical aspects of climate change remains arduous, the identified large-scale IVT regimes and
upstream flow analyses can offer useful references for assessing dynamical changes in the future. Here, we use them to
evaluate the moisture dynamical environments simulated by TaiESM1. Figure 7a–b displays the large-scale IVT regime
frequency based on the ERA5 data and the TaiESM1 historical simulation, respectively. The frequency maps suggest that
TaiESM1 reasonably simulates the circulation state over the AAM domain, further supported by regime composites (Fig.
S10). Under the SSP5-8.5 scenario, TaiESM1 projects distinct changes in the regime frequency at the end of the century (Fig.
7c), including an overall increase in the Category C patterns (mainly involving Western North Pacific tropical cyclones, Fig.
3c and S2), despite the slight underestimations of this category in the historical period. Evaluating established weather
regimes through historical and future datasets are common in statistical downscaling studies to preliminarily examine the
changes in identified large-scale circulation states (Zappa, 2019) and should provide guidance for the choice of GCMs for
further study (Maraun et al., 2017). Particularly, the trained VAE with individually-normalized IVT fields addresses this step
efficiently with emphasis on spatial structure over intensity discrepancies.

Critical moisture transport direction and intensity can be focused during upstream analyses. Similarly conditioned on the six
heavy-rainfall-promoting large-scale IVT regimes (C3, C5, B1, B2, B4, B5) for southwestern Taiwan, Fig. 7d–e also
suggests that TaiESM1 reasonably simulates the upstream IVT conditions, albeit slightly weaker and with more southerly





components. Compared to the thermodynamic influence from a vertically uniform 3K warming—derived from scaling historical simulations with the Clausius-Clapeyron relation (Fig. 7f), TaiESM1 projects intense southwesterly moisture transport to be more common under the SSP5-8.5 scenario (Fig. 7g). The increase of the most intense IVT (≥500 kg/m/s) in the southwesterly direction is much larger than what the thermodynamic response predicts (Fig. 7f), suggesting contributions from increased wind speed and altered wind direction frequency. In other words, environments favouring upstream-inflow-

dominant heavy rainfalls over southwestern Taiwan are expected to become more frequent according to TaiESM1 projections.

Evaluating crucial dynamical regimes in various GCM simulations can help explore the dynamical changes, which can be treated as different storylines, and the storyline-dependent local weather projections can then be obtained using reliable high-resolution models (Shepherd, 2016; Shepherd, 2019). With the conventional, unconditional probability estimates factorized

into several storyline estimates (conditioned on different dynamical states) (Shepherd, 2019), future occurrences of heavy rainfall events above certain thresholds can also be articulated with better-bounded uncertainties. The components of our process-based framework—namely the VAE, the upstream analyses, and TaiwanVVM—possess potential to informative local projections in this regard.

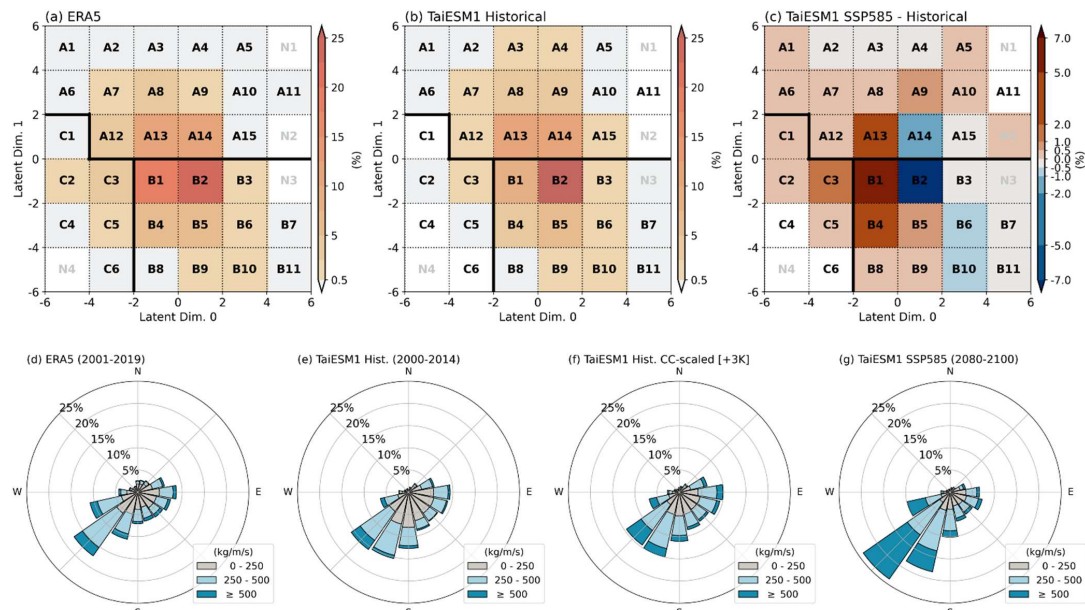

**Figure 7: Historical and future assessments of large-scale and upstream IVT with TaiESM1. (a)(b) show large-scale IVT regime frequency via ERA5 and TaiESM1 historical datasets. (c) shows the frequency differences between TaiESM1 SSP5-8.5 projection (2080–2100) and the historical period. (d)(e) show upstream IVT distribution via ERA5 and TaiESM1 historical datasets. (f) are**



**thermally-scaled from (e) according to the CC-equation with a vertically-uniform 3K warming in 1000–700 hPa. (g) shows the upstream IVT distribution of TaiESM1 SSP5-8.5 projection (2080–2100).**

### 4.2 Applicability to other AAM areas

We explore whether the identified large-scale IVT regimes (Fig. 3) and the upstream IVT analyses (Fig. 4) can serve as general indicators—for quantifying critical dynamical environments to local heavy rainfalls—in other topographically-complex areas within the AAM region. We chose the Western Ghats, the eastern coast of the Bay of Bengal, western Luzon, and southern New Guinea (Fig. 8a) as examples, as they possess similar coast-to-mountain topography to that of southwestern Taiwan. Despite different scales and detailed terrains features, the topography in these areas plays a huge role in shaping the sensitive heavy rainfall responses to upstream conditions (Chang et al., 2005; Shige et al., 2017; Riley Dellaripa et al., 2020; Mohandas et al., 2020; Krishna et al., 2021; Nicholas and Boos, 2024). Here we demonstrate with the area-average daily precipitation from IMERG (similar to Fig. 4b, d), where the dominant driver distinction, such as local circulation and upstream inflow, can be a topic of further study.

The distribution of heavy rainfalls over the large-scale IVT regimes exhibit distinct characteristics in these four areas (Fig. 8b–e). For the three areas in the Northern Hemisphere, heavy rainfalls in boreal summer occur predominantly in the Category B regimes (Fig. 8b–d) featuring abundant moisture transport from South Asian monsoon westerlies and moist vortices, which are also commonly-described patterns in seasonal-averaged or extreme rainfall event studies (e.g., Chang et al., 2005; Mohandas et al., 2020). In southern New Guinea, heavy rainfalls can be promoted in both Category A and B regimes, corresponding to their reported weaker seasonality compared to other areas (Chang et al., 2005; Biasutti et al., 2012; As-syakur et al., 2016) and stronger association with the incoming southeasterly flow (Fig. 8j). In western Luzon, heavy rainfalls are frequently affected by tropical cyclones over the Western North Pacific and the South China Sea (Bagtasa 2017; 2019) in addition to the prevailing monsoon flow (Fig. 8d). Susceptibility to various systems from adjacent oceans contributes to higher IVT variability upstream of western Luzon (Fig. 8h), which is reminiscent of IVT conditions upstream of southwestern Taiwan (Fig. 4d), compared to the Western Ghats and the eastern coast of the Bay of Bengal (Fig. 8f, g).

Figure 8 is an illustration of concept of how this downscaling strategy based on multi-scale moisture dynamical control can work in other areas, where environment–topography interaction strongly influences local heavy rainfall outcomes. We expect the large-scale IVT regimes and the upstream IVT analyses to efficiently narrow down the rainfall-relevant dynamical regimes, where high-resolution modelling tools and in situ observations can help unravel local rainfall responses in fine spatial scales.



**Figure 8:** Demonstration of the multi-scale IVT analyses linking regional rainfall in other areas. (a) Rainbow shading denotes the average daily precipitation in the warm season. Brownish shading illustrates the topography. Blue crosses denote the regions corresponding to (b)–(j), and blue dashed boxes show the chosen upstream. Southwestern Taiwan is marked by a black box. (b)–(e) Latent space with daily samples marked by regional-average daily precipitation. Annotated regimes are chosen visually based on



the distribution of heavy rainfall events. (f)–(j) Regional-average rainfall distributed against upstream IVT conditions, conditioned on the marked large-scale IVT regimes in (b)–(e).

### 4.3 Mechanisms of distinct rainfall behaviours over Taiwan's complex topography

In the past, local-circulation-dominant weathers are oftentimes intuitively linked to "drier environments" surrounding Taiwan, since the heavily precipitated area is usually very localized (i.e., small areal fraction, such as Fig. 5a–b). Boxplots of 925 hPa specific humidity (bottom panels in Fig. 6a, d), however, show overlapping moisture ranges throughout the low-levels between local-circulation and upstream-inflow dominant cases (Fig. S11) in the upstream region of Taiwan (Fig. 1b). The similar moisture conditions in the surrounding environment suggest that in the six heavy-rainfall-promoting large-scale IVT regimes (Fig. 4a–b), the disparate rainfall characteristics over Taiwan stem more from upstream wind dynamics—including both wind direction and wind speed—than from moisture availability. The local-circulation-dominant weathers are thus more akin to "inefficient moisture transport from the environment" than "drier environments."

In both tropical observation studies and idealized numerical experiments with more uniform wind directions (Smith et al., 2012; Shige et al., 2017; Wang and Sobel, 2017; Aoki and Shige, 2024), wind speed is found to control the shift from thermally-driven to mechanically-driven moist convection (Kirchbaum et al., 2018). Two of the reported rainfall characteristics accompanying the shift of driving mechanism are similarly observed over the complex terrain of southwestern Taiwan: the nonmonotonic response of rainfall to wind speed and the diminished diurnal structure of mechanically-driven rainfall. Figure 9a shows the area-average daily rainfall ($A$) on days with upstream IVT direction within ($230° \pm 2.5°$), distributed against the upstream wind speed and specific humidity at 925 hPa. It shows that rainfall increases more significantly with respect to wind speed than moisture, which even exhibits an abrupt jump with wind speed exceeding $10\ m/s$. Figure 9b shows the composites of hourly rainfall over southwestern Taiwan across all local-circulation and upstream-inflow dominant cases. The local-circulation dominant cases exhibit a clearer rainfall peak in the afternoon, compared to the upstream-inflow cases, but the latter generally results in greater amount of rainfall. Notably, since we study the summertime convection over a tropical topographic area, intense convection under upstream-inflow dominant weathers is likely triggered jointly by thermal and mechanical forcings. Further investigation into how the destabilizing processes from surface heating and moist inflow shape local instability might shed light on the variability of upstream-inflow-dominant heavy rainfall events regarding location (Fig. 6c–e) and even duration that bear disastrous impacts.

As these detailed physical processes within the upstream–local scale are embedded in and greatly modulated by the regional circulations states, we further highlight the theoretical grounds recently developed to elucidate the role of background moisture gradient, cross-slope wind, and large-scale advection in producing strong mechanically-driven precipitation over topography (Nicholas and Boos, 2022; 2024; 2025). How these processes across scales are represented in our current consideration of upstream and large-scale moisture dynamics, how to further inquire their real-world precipitative influence via semi-realistic LES modelling such as TaiwanVVM, and whether they help justify the adequacy of local areas for applying the proposed downscaling strategy (Fig. 8) might be key extensions of this work.



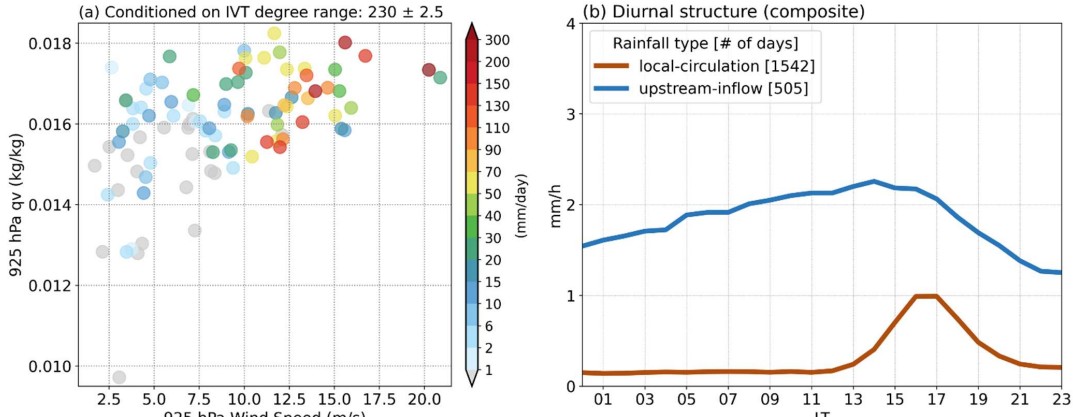

**Figure 9: Two observed features of southwestern Taiwan area-average rainfall. (a) Rainfall values on day with upstream mean IVT in $230° \pm 2.5°$, distributed according to the wind speed (x-axis) and specific humidity (y-axis) at 925 hPa at the upstream spot $(21.5° N, 118.5° E)$. (b) Hourly rainfall of local-circulation and upstream-inflow dominant cases, defined as in Fig. 4d.**

## 5. Conclusion

In this study, we take heavy rainfalls over Taiwan's complex topography as an example to investigate a process-based framework that translates crucial moisture dynamical environments to local rainfall over the challenging AAM region, which possesses promising potential as an alternative to conventional downscaling approaches. The VAE represents daily large-scale IVT regimes by efficiently extracting the multi-scale spatial features and can be applied to unseen datasets such as GCM simulations. The upstream IVT analyses allows detailed forcing types for local heavy rainfall to be identified, which guide the choice of representative environmental profiles. Finally, the semi-realistic large-eddy simulations explicitly realize the rainfall outcomes resulted from environment–topography interactions. When applying to future projections, the identified dynamical control serves as the conditioning basis for the thermodynamic and dynamical responses, and the different levels of uncertainty that accompany, to be separately addressed. This pathway towards future projections of local rainfall extremes reduces the mingled uncertainties inherent in conventional approaches, which, therefore, bears potential to areas also experiencing frequent topographic heavy rainfalls within the AAM region.

## 6. Code and data availability

Code and software are archived at https://doi.org/10.5281/zenodo.17230674 (Guo, 2025a) along with processed data of smaller sizes. Larger data files are archived at https://doi.org/10.5281/zenodo.17199183 (Guo, 2025b).

Original datasets employed in this study are accessed via the following sites:



1. ERA5: Pressure level data at https://cds.climate.copernicus.eu/datasets/derived-era5-single-levels-daily-statistics?tab=download and land surface data at https://cds.climate.copernicus.eu/datasets/reanalysis-era5-land?tab=overview, provided by Copernicus Climate Change Service information. Neither the European Commission nor ECMWF is responsible for any use that may be made of the Copernicus information or data it contains.

2. IMERG: https://disc.gsfc.nasa.gov/, provided by Goddard Earth Sciences Data and Information Services Center (GES DISC).

3. TaiESM1: historical run at https://ipcc-browser.ipcc-data.org/browser/dataset/7162/0 and SSP5-8.5 run at https://ipcc-browser.ipcc-data.org/browser/dataset/6712/0, provided by World Data Center for Climate at DKRZ.

4. IBTrACS: https://www.ncei.noaa.gov/products/international-best-track-archive, provided by National Centers for Environmental Information, NOAA.

5. Central Weather Administration observation data: https://asrad.pccu.edu.tw/, provided by Atmospheric Science Research and Application Databank (ASRAD).

TaiwanVVM simulation data are available from the corresponding author upon reasonable request.

## 7. Author contribution

JXG designed the VAE model details and performed the model training. JXG, WTC, and CMW designed the TaiwanVVM experiments, and CMW performed the simulations. JXG developed the code for analyzing and displaying the results. JXG and WTC prepared the manuscript with contributions from all co-authors. All authors reviewed and approved the manuscript.

## 8. Competing interests

The authors declare that they have no conflict of interest.

## 9. Acknowledgements

The authors sincerely thank the National Center for High-performance Computing (NCHC) and Central Weather Administration (CWA) for providing the high-performance computation platform to conduct the simulations.

## 10. Financial support

This work was supported by the National Science and Technology Council in Taiwan through grants NSTC114-2119-M-002-025-, NSTC114-2111-M-002-001- (J.-X. Guo and W.-T. Chen) and NSTC114-2123-M-002-012-, NSTC114-2119-M-002-027-, NSTC113-2124-M-002-015- (C.-M. Wu) to National Taiwan University and National Taiwan University Grants NTU-114L7819 (W.-T. Chen).



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
