# Peer review of "A Promising Downscaling Strategy for Topographic Heavy Rainfalls over the Asian-Australian Monsoon Region by Leveraging Multi-Scale Moisture Dynamical Control"

_EGUsphere, 2025_

## Referee Comment (RC1)

The manuscript presents a clear, innovative, and highly relevant approach to addressing critical challenges in projecting extreme rainfall in tropical regions with complex topography. The integration of machine learning Variational Autoencode (VAE) with high-resolution large-eddy simulations (TaiwanVVM) to develop a process-based downscaling framework constitutes a valuable methodological contribution. The main argument of this study disentangling thermodynamic forcing from dynamics and demonstrating that upstream wind dynamics serve as the primary control of heavy rainfall types. It is supported by detailed multi-scale analyses.

Overall, the study demonstrates strong potential and offers important insights for the downscaling of topographic heavy rainfall. However, several issues require clarification and revision to enhance the clarity, reproducibility, and interpretability of the manuscript. My specific comments are as follows:

- The objective of this study is to propose a downscaling method for Topographic Heavy Rainfalls over the Asian–Australian Monsoon Region by leveraging multiscale moisture dynamical control. However, the manuscript provides little information regarding the resulting downscaled resolution and its accuracy relative to topographic rainfall. This omission makes it difficult for readers to assess how the proposed method performs compared with existing GCM-based downscaling approaches.
- Regarding Figure S1, it is unclear why the spatial differences between IMERG and CWA appear so large. Previous studies using gridded CWB datasets show more consistent spatial patterns between IMERG and CWB.
  Reference: https://www.sciencedirect.com/science/article/pii/S0169809518304666
- 3. The introduction should elaborate on why Integrated Vapor Transport (IVT) is important in the downscaling process.
- 4. Although the VAE-based method shows improved correlation and reduced RMSE, the differences compared with EOF appear relatively modest (Figure S5). Have you considered comparing the clustering results between VAE and EOF?
- 5. The organization of figures in both the main text and the supplementary material is at times confusing. Several figures are referenced repeatedly across the Methods and Results sections, making it difficult for readers to follow—particularly in Section 2.2 (Methods) and Section 3.1 (Results). In addition, referring to certain figures in both the Results/Discussion and the Methods (e.g., line 141) does not enhance clarity and forces readers to jump back and forth.

---

## Author Comment (AC1)

**A Promising Downscaling Strategy for Topographic Heavy Rainfalls over the Asian-Australian Monsoon Region by Leveraging Multi-Scale Moisture Dynamical Control**

Jia-Xin Guo[1], Wei-Ting Chen[1], and Chien-Ming Wu[1]

[1] Department of Atmospheric Sciences, National Taiwan University, Taipei, 106319, Taiwan

The authors sincerely thank the two anonymous reviewers for their constructive feedback, which has significantly strengthened the manuscript. We have carefully revised the paper to address these comments. The primary modifications include:

1. Revised the Introduction to better situate our strategy within the "storyline approach", elaborate on the importance of IVT, and highlight the downscaled resolution of precipitation outcomes.
2. Added clustering comparisons to justify the selection of VAE over EOF for regime identification.
3. Improved Figure 3 to emphasize the physical interpretation of machine-learning-based regime identification.
4. Included pattern correlation analysis between observed and simulated rainfall composites to quantitatively validate simulation performance, with updates to Figure 6.
5. Clarified key concepts including "VAE approach", "large-scale regime", "dominant drivers", and resolved ambiguities regarding the downscaling strategy.
6. Ensured consistency between rain gauge data and the 1-km gridded observational rainfall.
7. Corrected typographical and grammatical errors throughout the manuscript.

Please find our point-by-point response to the reviewers' comments below. Our responses are marked in blue, and modifications to the main text and Supplementary Material are highlighted in **bold**. Page (P) and line (L) numbers refer to the revised manuscript.

**Referee #1**

1. The objective of this study is to propose a downscaling method for Topographic Heavy Rainfalls over the Asian–Australian Monsoon Region by leveraging multiscale moisture dynamical control. However, the manuscript provides little information regarding the resulting downscaled resolution and its accuracy relative to topographic rainfall. This omission makes it difficult for readers to assess how the proposed method performs compared with existing GCM-based downscaling approaches.

   We appreciate the Reviewer for highlighting this critical point.

   First, we use TaiwanVVM to render the local rainfall outcomes at 500 m spatial resolution, which is the final step of this downscaling strategy. **We added this point in the Abstract (P1, L12) and Introduction (P2, L52–55) to highlight it, and on the latter, we added that the resolution is much finer than typical RCM-based dynamical downscaling results of O(10km). We also revised P,14 L337–340 to note that the proposed strategy is particularly promising for regions with high-resolution models resolving critical terrain-related processes.**

Second, we agree that some form of metrics will provide more direct assessments of our precipitation simulation results. Since the downscaling strategy aims at providing interpretable projections for the two distinct rainfall types (local-circulation/upstream-inflow dominant), **we added pattern correlation between the composite patterns of observation and TaiwanVVM simulation of the chosen ~30 cases (modified Figure 6).** The correlation is calculated by sampling the rainfall values from TaiwanVVM simulation on grids closest to the rain gauge positions. The correlation values exceed 0.6 and 0.7 for the simulated local-circulation and upstream-inflow dominant cases, respectively, and remain high when compositing all samples within the rainfall types (0.55 and 0.68), which suggests qualitative pattern agreement. **We revised P13, L3217–328 to provide more thorough assessments of the similarities and discrepancies between the observed and simulated rainfall patterns.**

2. Regarding Figure S1, it is unclear why the spatial differences between IMERG and CWA appear so large. Previous studies using gridded CWB datasets show more consistent spatial patterns between IMERG and CWB.

   Reference: https://www.sciencedirect.com/science/article/pii/S0169809518304666

   We thank the Reviewer for pointing out this reference (Huang et al., 2018, hereafter H18). H18 is cited in our submitted manuscript, describing the difficulty of IMERG data in representing the precipitation around the southern mountains of Taiwan. Below, we reproduced our Fig. S1 using identical color scales with H18's Fig. 2 to ensure consistency (Fig. R1). We additionally added a version using gridded rainfall dataset from Taiwan Climate Change Projection Information and Adaptation Knowledge Platform (TCCIP) at 1 km resolution for reference. We note that our Fig. S1 shows the average through Apr–Sep in 2001–2019, while H18's Fig. 2 shows 3-month averages in 2014–2017. We thus compared the rainfall pattern in our Fig. S1 to the combined pattern of MAM and JJA in H18's Fig. 2.

   Our Fig. S1 is very similar to H18's Fig. 2 when using identical color scales, which shows the largest discrepancy near the mountain ranges from the mid-section towards southwestern. In constrast, IMERG data is challenged to capture rainfall larger than 12 mm/day. Based on the consistency, the higher rainfall levels (20 and 25 mm/day) in our original Fig. S1 further highlight the underestimation over the mountains. Due to our desired emphasis on the challenges posed by complex terrain, we decided to retain the original version in the Supplementary.

[Figure]

**Figure R1: Using the color scales of H18's Figure 2 (left column) to draw our Figure S1 (middle column). The right column shows higher color scales to emphasize the rainfall discrepancy over the mountain ranges.**

3. The introduction should elaborate on why Integrated Vapor Transport (IVT) is important in the downscaling process.

   We thank the Reviewer for highlighting this critical revision. **We modified the entire second paragraph of Introduction to elaborate on our reasoning for choosing IVT and the importance of founding the downscaling strategy on multi-scale moisture dynamics (P2, L31–43).**

4. Although the VAE-based method shows improved correlation and reduced RMSE, the differences compared with EOF appear relatively modest (Figure S5). Have you considered comparing the clustering results between VAE and EOF?

   Yes, **we revised the Supplementary to focus the comparison between VAE and EOF in Fig. S2– 4.** The PCA is fitted with the training subset and used to transform the entire dataset. The input data is identical to those used in VAE (individually-normalized daily IVT intensity fields).

   The overall clustering results between VAE and EOF appears relatively similar (Fig. S3), since EOF mostly captures larger structures. The main differences appear in the regional details that shape the prominent cross-scale interaction within the AAM circulation patterns, as we originally used Fig. S4 to demonstrate (**now moved to Fig. S2b**). EOF is especially challenged to represent (1) the monsoon pattern with extended IVT into the Western North Pacific, and (2) the TC-related patterns. **We added Fig. S3 and S4 to highlight that this leads to less ideal classification of regimes.** Since these detailed regional flow structures are critical to topographic heavy rainfalls, we chose VAE over EOF for regime identification. **We modified P5, L119–124, accordingly.**

5. The organization of figures in both the main text and the supplementary material is at times confusing. Several figures are referenced repeatedly across the Methods and Results sections, making it difficult for readers to follow—particularly in Section 2.2 (Methods) and Section 3.1(Results). In addition, referring to certain figures in both the Results/Discussion and the Methods (e.g., line 141) does not enhance clarity and forces readers to jump back and forth.

   We appreciate this constructive comment. **We have edited out repetitive figure references**

throughout the main text, especially in Section 2.2. We also rearranged the figures in the Supplementary to ensure a smoother flow.

Data availability: The 1 km gridded precipitation data are available from the Taiwan Climate Change Projection Information and Adaptation Knowledge Platform (TCCIP, https://tccip.ncdr.nat.gov.tw).

**Referee #2**

1. **Incorrect calculation of IVT.**

   I am a little concerned by the authors' definition of IVT. IVT is a vector field but it is written as if it is a scalar field. The written equation (Eq 1, pg 4) is incorrect. The scalar IVT should be the magnitude of IVT. I.e.

   $$\left(IVT_x, IVT_y\right) = (-\frac{1}{g}\int_{1000}^{700} qu\, dP, -\frac{1}{g}\int_{1000}^{700} qv\, dP)$$

   $$IVT = \sqrt{IVT_x^2 + IVT_y^2}$$

   Where IVTx and IVTy are the x and y components of the (vector) IVT field, u and v are the x and y components of the velocity field, q is the specific heat, g is the acceleration due to gravity and P is the pressure.

   We appreciate the Reviewer for emphasizing the correct formulation of IVT, which is exactly how we calculated it. We used the notation similar to Rutz et al. (2014) **(added this citation in P2, L38)** and realized that absent vector annotation on the left-hand side can cause confusion. **Accordingly, we revised Eq. (1) as**

   $$\overrightarrow{IVT} = (IVT_x, IVT_y) = -\frac{1}{g}\int_{1000}^{700} q \cdot \vec{v}\, dP, \tag{1}$$

   **and added Eq. (2) to indicate the calculation of IVT intensity:**

   $$\left|\overrightarrow{IVT}\right| = \sqrt{IVT_x^2 + IVT_y^2}. \tag{2}$$

   **These modifications are in P3, L83–86.**

   To further address the Reviewer's concern, we restate our use of IVT in this study as follows: The large-scale patterns are identified via VAE using only the IVT magnitude (i.e., $\left|\overrightarrow{IVT}\right|$) fields, which we refer to as "IVT intensity" in the main text. The upstream analyses are based on the IVT vector (i.e., $\overrightarrow{IVT}$), which we discuss both the direction and the intensity. The reason why we only use $\left|\overrightarrow{IVT}\right|$ to identify large-scale patterns is to constrain the number of identified regimes, and the direction information is explicitly investigated during upstream flow analyses. **We revised P5, L132–134 and P7, 176–179 to state this more clearly. We also added notations of $\left|\overrightarrow{IVT}\right|$ and $\overrightarrow{IVT}$ in many places throughout the main text to remind the readers.**

2. **Added value of machine learning approach is not clear.**

   The title of the paper claims that 'multiscale moisture dynamical control' is leveraged providing a

'promising downscaling strategy' for topographic heavy rainfall. However, despite the VAE approach, the downscaling ultimately boils down to two pre-known drivers (local-circulation vs. upstream-inflow; Sec. 3.2–3.3; Fig. 4–6). The upstream in-flow regime is well documented, as the authors state. The spatial structure of IVT could be found simply by compositing IVT during days dominated by upstream-inflow. The VAE methodology seems unnecessary here and ultimately identifies only one relevant large-scale regime (upstream inflow). The difference between the EOF method and the VAE method on the test data in S5, does not look significant.

To address all the concerns mentioned in this comment, we will clarify (1) the confusion between large-scale regimes and dominant drivers, (2) the purpose of using VAE, and (3) the added value of using VAE.

First, we clarify that the "large-scale regimes" refer to the weather system configurations over the Asian-Australian monsoon region (Fig. 1a), whereas the "dominant drivers" refer to the local-circulation/upstream-inflow (using southwestern Taiwan as the focused example), indicated by upstream $\overrightarrow{IVT}$ conditions (Fig. 1b). In this study, we establish daily large-scale IVT regimes to identify heavy-rainfall-promoting large-scale patterns (Fig. 4a–b), constrain the regional circulation state for building storylines, and evaluate the simulation of GCMs (Fig. 7). Dominant drivers are indicated by the upstream $\overrightarrow{IVT}$, which guides the choice of the representative environmental profile for the LES. In Section 3.2, we identify six relevant large-scale regimes for promoting heavy rainfalls over southwestern Taiwan (Fig. 4a–b). Under these large-scale weather patterns, heavy rainfalls are predominantly driven by upstream inflow when the upstream $\overrightarrow{IVT}$ corresponds to $200° \leq direction \leq 280°$ and $intensity \geq 250 \frac{kg}{m \cdot s}$ and by local circulation otherwise (Fig. 4c–d). Both local circulation and upstream inflow are relevant drivers that can induce daily rainfall over steep topography more than 100 mm/day in distinct patterns (Fig. 5). **We revised P11, L268–270 to highlight this point.**

Second, we restate that the purpose of using VAE is to objectively extract daily large-scale IVT regimes from daily patterns. Our proposed downscaling strategy aims to provide interpretable future projections following the storyline approach, which projects plausible futures conditioned on distinct large-scale states and thereby constrain the uncertainties from the uncertain response of regional circulation. **The Introduction is revised to make this point clearer (P1, L19–43).** Even when building plausible storylines for the rainfall type dominated by upstream inflow, it is critical to identify what large-scale state modulates the strong upstream inflow. For example, stronger upstream inflow for southwestern Taiwan can be induced by either the monsoon circulation variability or a nearby TC east off Taiwan. These two scenarios represent different large-scale states, and hence, different storylines, for topographical rainfall over southwestern Taiwan, despite that their upstream IVT can be similar. Compositing IVT during days dominated by upstream-inflow, as the Reviewer suggested, is likely to merge these different scenarios of large-scale patterns, hampering the chance to delineate their respective contribution to current and future rainfall in southwestern Taiwan. **We revised P11, L284–289 to highlight this point.**

Third, the added value of VAE, especially compared to EOF, mainly lies in regional details that shape

the prominent cross-scale interaction within the AAM circulation patterns, as we originally used Fig. S4 to demonstrate (**now moved to Fig. S2b**). EOF is especially challenged to represent (1) the monsoon pattern with extended IVT into the Western North Pacific, and (2) the TC-related patterns. **We added Fig. S3 and S4 to highlight that this leads to less ideal classification of regimes. We rearranged Fig. S2–S4 for comparing VAE and EOF's performance and summarized these points in P5, L119–124.**

3. **Lack of validation of LES.**

   LES evaluation uses ~30 cases per type (Fig. 6, page 15 and ll 303, page 13). It is claimed that there is a qualitative pattern agreement but from examining Fig 6, it looks like the results from LES don't agree well with the observations. Both 6b and 6e (observed) look different compared to 6c and 6f (LES model). No formal skill metrics (e.g. spatial correlation) are provided..

   We thank the Reviewer for pointing out the need for formal skill metrics. As suggested, **we modified Figure 6 by adding the observational composites of those simulated cases (now Fig. 6c and 6g) and their pattern correlations with the simulation**. The correlation is calculated by sampling the TaiwanVVM-simulated rainfall on grids closest to the rain gauge positions, as demonstrated below (Fig. R2). The correlation values exceed 0.6 and 0.7 for the simulated local-circulation and upstream-inflow dominant cases, respectively, and remain high when compositing all samples within the rainfall types (0.55 and 0.68), which suggests qualitative pattern agreement. **We revised P13, L317–328 to assess more thoroughly the similarities and discrepancies between the observed and simulated rainfall patterns.**

[Figure]

**Figure R2: Calculate pattern correlation between composites by sampling TaiwanVVM-simulated rainfall at grids closest to the rain gauges. In each panel title, the numbers in paratheses indicate the number of composite cases, matching those of Figure 6.**

4. **Rainfall over complex topography is not investigated.**

The manuscript cites complex topography as the motivation for the framework, yet later sections refer only to 'localised' vs. 'widespread' rainfall without explicitly linking these patterns to terrain-driven processes. LES simulations could have been used to quantify orographic lifting, slope orientation effects, and land–sea breeze interactions. Indeed, rain gauges on complex terrain are less reliable since they are difficult to access and are generally sparser. Analysis of this sort would be of scientific interest. Without explicit topographic diagnostics (e.g., upslope vs. lee-side differences), the study does not fulfil its stated goal of providing interpretable projections for complex terrain.

We absolutely agree with this comment about the potential of investigating more detailed terrain-driven processes using LES. Indeed, the literature cited in our text (**P6, 158–159**) are previous investigations dedicated to the terrain-related processes and the resultant intense convection using VVM (idealized topography) and TaiwanVVM (realistic topography). These studies underly our confidence in using TaiwanVVM for rendering local rainfall outcomes, knowing that explored convective processes are reliable and further investigation is possible.

However, we emphasize that the full interpretability of rainfall projection through the proposed

downscaling framework lies in the holistic understandings covering the large-scale moisture dynamical environments, the choice of representative environmental profiles, and the explicit representations of the terrain- and convection-related processes by the LES to produce rainfall outcomes. It aligns with the storyline approach and is a contrast to downscaling unconditionally from GCM ensembles with RCMs, which introduces mingled uncertainties and diminishes interpretability. **The Introduction is revised to strengthen this point (P1, L19–43).**

As discussed in Section 4.3, we are very motivated to investigate the terrain-related processes under local-circulation and upstream-inflow dominant weathers, but they are currently beyond the scope of this study.

5. **Incorrect or unclear calculation of rainfall metrics (R and A).**

   The study uses metrics R (ratio of 'area summed' rainfall over Taiwan compared to that of southwest Taiwan) and A the area averaged rainfall over southwest Taiwan (Eq. 3 and Eq. 4, respectively, page 10). It is unclear what 'area summed' means in the calculation of R and A. Furthermore, it is unclear what A is measuring. A (defined by Eq. 4) appears to estimate the regional rainfall intensity by summing raw rain gauge totals and dividing by gauge count for averages. This is incorrect. Furthermore, if R does faithfully provide a measure of how localised the rainfall is, Fig 4c seems to show that there is no relationship between R and IVT intensity or direction.

   The Reviewer expressed concerns about three points in this comment: [1] the meaning of area-summed daily rainfall ($S$ in Eq. 3, now Eq. 4 in the revised version) [2] incorrect definition of regional average rainfall A with rain gauge data, and [3] no relationship between R and upstream $\overline{IVT}$. We respond to each point as follows.

   [1] The area-summed daily rainfall ($S$) is defined for the purpose of estimating how much of the rainfall over the Taiwan Island concentrates over southwestern Taiwan on a specific day (i.e. $S$ is an intermediate step for obtaining the metric "$R$"). For simplicity, $S$ is calculated by summing the daily rainfall amounts measured by all rain gauges within the specified area. This allows us to straightforwardly compare the rainfall received by the southwestern area and the entire island by taking their ratio. **We revised P10, L240–249 to make this point clearer.**

   [2] As the Reviewer's understanding, $A$ measures the regional daily rainfall by dividing $S$ with the gauge counts. The estimation using rain gauge data is credible owing to the dense rain gauge network over Taiwan, especially southwestern Taiwan. To address the Reviewer's concern, we reproduced Fig. 4 with the gridded observational rainfall data from Taiwan Climate Change Projection Information and Adaptation Knowledge Platform (TCCIP) at 1 km resolution and compared it with the original Fig. 4 (Fig. R3). The two versions appear very similar, especially the southwestern average (i.e., the metric $A$), with only slightly higher southwestern rainfall ratio (i.e., the metric $R$) when using the gridded TCCIP rainfall. This ensures consistency with our original results. In addition to the agreement between both versions, the rain gauge data aligns better with our focus on the high-resolution rainfall patterns over complex topography. In Fig. R4, we further demonstrate how the rain gauge data can better capture the highly-localized rainfall hotspots than the gridded dataset. **We added this point in P4, L96–97.**

   [3] Following point [1], we clarify that $R$ does not measure how localized the rainfall is, but

measures whether rainfall concentrates over southwestern Taiwan on that day. It has to be considered jointly with $A$, the area-average rainfall, to imply whether the rainfall pattern is more likely to be localized (low $A$) or widespread (high $A$). As the Reviewer expected, which is also exactly our point, high $R$ does not show a specific structure respecting upstream $\overrightarrow{IVT}$. This suggests that rainfall concentrated over southwestern Taiwan can happen on days with weak or strong incoming flow. In contrast, high $A$ exhibits a clear structure clustered towards stronger southwesterly $\overrightarrow{IVT}$. Juxtaposing Fig. 4c and 4d thus reveals that when heavy rainfalls emerge over southwestern Taiwan, the rainfall pattern is likely to be localized under weak incoming flow and more widespread under strong incoming flow. **We modified P10, L246–249 and P11, L263– 267 to strengthen this point.**

[Figure]

**Figure R3: Comparison of Figure 4 using different rainfall datasets. The original version (left) is based on rain gauge data, whereas the reproduced version (right) is based on the gridded rainfall data at 1 km resolution from TCCIP.**

[Figure]

**Figure R4: Comparison of local-circulation-dominant rainfall patterns between rain gauge data and gridded rainfall data to demonstrate their representation of localized rainfall hotspots. The rain gauge panels are identical to Figure 6b–c in the main text.**

6. **Over-reliance on IVT-only for regime definition.**

The VAE is trained solely on IVT maps (Sec. 2.2), after daily min–max normalization and coarsening (see point 8). Two environments with similar IVT can produce very different rainfall outcomes. This use of IVT alone is not adequately justified.

We agree with the Reviewer that two environments with similar IVT (whether in the sense of large-scale pattern or upstream condition) can produce very different rainfall outcomes, and this is exactly why the semi-realistic LES component is essential to the downscaling strategy. When initializing the TaiwanVVM LESs, the vertical profiles of temperature, specific humidity, and horizontal winds are given to represent the environmental forcing (P6, L165–169). Therefore, even on days sharing similar IVT environments, TaiwanVVM can simulate the detailed precipitation outcomes over topography corresponding to the respective dynamic and thermodynamic environment. This point is nicely demonstrated by the simulated rainfall variabilities for the two distinct rainfall types (Fig. 5). We emphasize that this framework is not intended to infer local rainfall outcomes solely based on the IVT environments. Instead, the IVT investigation helps us identify the representative environment, and we use it with a model resolving the crucial processes to enhance interpretability of the rendered rainfall outcomes. **This point underlies Section 3.3, and we additionally revised P2, L53–55 to place this point upfront.**

7. **Heuristic classification of regimes.**

Regime grouping (36 latent sub-cells divided into categories A, B and C) is based on visual inspection rather than an objective clustering appraoch (Sec. 3.1; Fig. 3). This undermines reproducibility and weakens claims of 'interpretable ML'. No quantitative mapping is provided between the latent coordinates and physically distinct processes beyond arbitrarily grouping different patterns in the IVT field.

To first clarify, our regimes are objectively classified by objectively discretizing the latent space. The three categories are then grouped based on visually inspecting all the classified daily IVT patterns, the regime composites, and their agreements with the regime representatives. Defining the

three categories is intended to enhance interpretability of the machine-learning results, by linking the objectively-identified regimes to our physical knowledge of domain weather patterns over the monsoon region. The three categories, therefore, involves a certain degree of subject interpretation, but it does not interfere the objective regimes. **We modified P7, 189–195 and moved Fig. S7 to Fig. 3b to strengthen this point for interpretable machine learning.** We note that our subsequent analyses of embedded upstream $\overrightarrow{IVT}$, local rainfall behaviors, and GCM evaluation are all based on the objectively-identified regimes, not the categories. It therefore does not affect the reproducibility of our results.

8. **Assumption of Gaussian statistics and issues with min-max normalisation.**

The VAE framework assumes IVT follows a Gaussian distribution (Sec. 2.2). IVT is non-Gaussian, tends to be heavy tailed with extreme values occurring more often than predicted by a Gaussian distribution. This assumption would underrepresent the extremes, which are the central process being investigated in this study. Furthermore, I am concerned about the daily min–max normalisation of IVT in Sec 2.2,Eq.2. This places low IVT days (that would have very little impact on the terrain driven rainfall) on equal footing with high IVT days. The authors own analysis in S6 shows that the normalisation is flawed. The normalisation should account for the variability in IVT intensity. The normalisation used also makes tropical-cyclone cores appear weaker, as the authors acknowledge in S6, this undermines the choice of defining category C as 'tropical cyclone related' (Sec. 3.1).

We appreciate this comment for us to clarify our methods.

First, the VAE framework does not assume IVT value follows a Gaussian distribution. The multivariate Gaussian distribution is introduced through the KL-divergence term in the loss function of VAE to regularize the training objective. It is meant to improve the dimension-reduction process compared to conventional autoencoders and does not assume the physical properties of the input samples. Imposing a multivariate Gaussian prior encourages meaningful distance in the latent space and smooth latent representations. Given our inputs as IVT patterns, this lets similar IVT patterns be clustered on the latent space more reasonably with smooth transition of circulation features. **We modified P4, L115–118 to make this point clearer.** Furthermore, we once again clarify that the heavy rainfalls focused in this study do not necessarily emerge under "extreme IVT" environments, whether in the sense of extreme "large-scale IVT pattern" or extreme "upstream IVT value". In fact, Fig. 4 and Fig. 8 demonstrate that upstream-inflow-dominant heavy rainfalls in the AAM region can emerge under common large-scale patterns and upstream IVT intensity starting from 250 kg/m/s. Not to mention the local-circulation-dominant heavy rainfalls can be triggered under weak incoming flow.

Second, the daily min–max normalization is performed to emphasize the daily structure of the large-scale IVT patterns, especially the cross-scale interaction induced by the relative positions and strengths between weather systems. Figure S6 shows that the weather system configuration on both days is similar—the monsoon system present over the Indian subcontinent, a tropical cyclone at the northeast of Luzon that is slightly stronger than the monsoon, and the enhanced flow induced by TC–monsoon interaction in the South China Sea. VAE is intended to classify these similar structures into the same regime, as they are expected to induce similar flow structures (over the South China

Sea in this example). **This point is explained in P5, L139–140 and reminded in P7, 185–188.** We agree the variability in IVT intensity should be considered, which is exactly why we explicitly investigate the upstream IVT intensity and direction in the local upstream next.

Third, Category C is determined as TC-related patterns through examining the daily IVT patterns in these regimes and the positions of historical tropical cyclones (originally shown in Fig. S2b, **now in Fig. S4a**), in addition to the reconstructed patterns. We find that almost all the days in Category C feature TCs in the Western North Pacific. We clarify once again that the individual normalization retains the daily structure of IVT patterns, including the structure of TCs. **We revised P8, L202–210 to clarify this point better.**

Data availability: The 1 km gridded precipitation data are available from the Taiwan Climate Change Projection Information and Adaptation Knowledge Platform (TCCIP, https://tccip.ncdr.nat.gov.tw).